# Immunogenicity and safety of a recombinant gE-Fc fusion protein subunit vaccine for herpes zoster in adults ≥50 years of age: a randomised, active-controlled, non-inferiority trial

The licensed adjuvanted recombinant glycoprotein E (gE) subunit vaccine (HZ/su) is highly effective against herpes zoster (HZ). This randomised, active-controlled, non-inferiority trial (ChiCTR2300079076) compared the immunogenicity and safety of a novel gE-Fc fusion protein vaccine candidate (LZ901) with HZ/su in 300 healthy adults aged ≥50 years without prior HZ vaccination in Wuxi, China. Participants received either two doses of LZ901 (30-day interval; n = 151) or HZ/su (60-day interval; n = 149). The primary outcomes was the proportion of participants with simultaneous positive responses to two or more cytokines (IFN-γ, IL-2, TNF-α, or CD40L) 30 days after the second dose (referred to as gE-specific CD4$^{2+}$/CD8$^{2+}$ T-cell responses). LZ901 demonstrated non-inferiority to HZ/su (margin > −10%) for both CD4$^+$ and CD8$^+$ T-cell responses. Significantly higher response rates were observed with LZ901 for CD4$^2$ + T-cell responses (83.0% [117/141] vs 58.1% [79/136]; p < 0.0001) and CD8$^2$ + T-cell responses (46.8% [66/141] vs 8.8% [12/136]; p < 0.0001). Adverse reactions were markedly lower with LZ901 (41.1% [62/151] vs 87.9% [131/149]; p < 0.0001), including grade 3 events (0.7% [1/151] vs 6.0% [9/149]). LZ901 induced superior cellular immunogenicity and exhibited a better safety profile than HZ/su in adults ≥50 years, supporting its potential as a promising HZ prevention candidate vaccine.

Herpes zoster (HZ), commonly known as shingles, results from the reactivation of the varicella-zoster virus (VZV), which establishes life-long latency in the dorsal root ganglia following primary infection during childhood[1,2]. Over 95% of adults aged 50 years or older have been infected with VZV, with an estimated lifetime HZ risk of 30%[2]. Postherpetic neuralgia (PHN), the most prevalent complication affecting 10–18% of HZ patients, manifests as chronic neuropathic pain that substantially diminishes quality of life[3,4]. Age-associated decline in

VZV-specific T-cell immunity drives exponential increases in HZ incidence[5], escalating from 4.0 to 4.5 cases per 1000 person-years in the general population to over 11.0 cases per 1000 person-years in individuals aged 80 years or older[6]. As global populations age, HZ has emerged as a growing health burden worldwide.

The HZ/su, a recombinant glycoprotein E (gE) subunit vaccine, consists of recombinant VZV glycoprotein E and a liposome-based AS01B adjuvant system, has demonstrated over 90% protection

e-mail: pengling@luzhubiotech.com; f.r.yan@163.com; changguili@aliyun.com; jingxin42102209@126.com

against HZ and PHN in adults aged 50 years and older, including those over 80 years of age, with sustained protection exceeding 70% over a decade[7–9]. Since HZ/su has the best reported protective efficacy to date to prevent HZ in individuals ≥50 years of age, it is globally recommended for preventing HZ. However, its liposomal AS01B adjuvant system contributes to substantial reactogenicity, with 10% of recipients experiencing grade 3 adverse reactions within 7 days post-vaccination[7]. Additionally, the high costs and limited production capacity of HZ/su compromised its accessibility, particularly in low- and middle-income countries (LMICs). These limitations necessitate the development of next-generation vaccines balancing immunogenicity, safety, and affordability.

LZ901, a novel gE-Fc fusion protein vaccine for HZ, employs a tetrameric structure combining four VZV gE domains with dual human IgG1 Fc fragments, adjuvanted with aluminum hydroxide[10]. Preclinical studies reveal that LZ901 is able to engage Fcγ receptors (FcγR) on antigen-presenting cells, resulting in enhancing cross-presentation and eliciting robust CD4[+] and CD8[+] T-cell responses alongside gE-specific antibodies[10]. Phase I/II trials (ChiCTR2200055617/ChiCTR220058609) confirmed its favorable safety profile and potent immunogenicity in adults aged 50–70 years. Currently, a multi-center, randomized, double-blind, placebo-controlled phase 3 clinical trial is ongoing in China to evaluate the efficacy, immunogenicity, and safety of LZ901 against HZ in adults (ChiCTR2300076253/ NCT06088745).

In order to accelerate the research and development process of LZ901, and to reveal its potential protective efficacy, we conducted a head-to-head clinical trial to demonstrate the non-inferiority of the immunogenicity of the LZ901 vaccine to the HZ/su vaccine. Here, we present the findings from this head-to-head comparison of the immunogenicity and safety of the LZ901 and HZ/su in adults aged 50 years and older in China.

## Results

### Trial population

Between December 25 and 29, 2023, a total of 357 participants were screened, of whom 301 were randomly allocated to receive either LZ901 vaccine or HZ/su vaccine. One participant in HZ/su group withdrew from the study before vaccination, and consequently 300 participants (151 in the LZ901 group and 149 in the HZ/su group) received at least one dose of the assigned vaccine and were included in the safety analysis. A total of 279 participants who completed two doses of vaccination and donated blood samples before and 30 days post vaccination were included in immunogenicity PPS analysis (Fig. 1). The mean age was -60.2 years in both groups, with 152 (50.7%) participants aged 50–59 years, 114 (38.0%) participants aged 60–69 years and 34 (11.3%) participants aged 70 years and older. 180 (60.0%) of 300 participants were female, and 296 (98.7%) of 300 participants identified as Han Chinese. Baseline characteristics of the participants were similar between the two groups (Table 1).

### Immunogenicity

The proportion of gE-specific CD4[2+] T-cell responders was higher in LZ901 participants than in HZ/su participants (83.0% (117/141) vs 58.1% (79/136)), $p < 0.0001$. The gE-specific CD8[2+] T cellular response followed a similar pattern, with 46.8% (66/141) of LZ901 group showing positive responses, compared to only 8.8% (12/136) in HZ/su group (Table 2). The proportion of CD4[2+] and CD8[2+] T-cell responders in LZ901 recipients differed from that in HZ/su recipients by 24.9% (95% CI: 14.5, 35.3) and 38.0% (95% CI: 28.5, 47.5), respectively. The lower bounds of the two-sided 95% CIs of these differences were both more than the predefined value −10%, and thus the non-inferiority criteria of the LZ901 vaccine were met compared with HZ/su vaccine. Of 4 cytokines, LZ901 induced higher response rates for IFN-γ, IL-2 than HZ/su (Fig. 2, Supplementary Table 1). Additionally, the proportions of LZ901 participants with simultaneous positive to 3 cytokines and 4

cytokines producing CD4[+] T-cell responses were higher than that of HZ/su participants, while that with simultaneous positive to 2 cytokines and 3 cytokines producing CD8[+] T-cell responses (Fig. 2 and Supplementary Table 2).

Both LZ901 and HZ/su vaccines significantly increased gE-specific CD4[+] T-cell cytokine production including IFN-γ, IL-2, TNF-α, and CD40L at 30 days after the second dose, compared to that at the baseline (all $p < 0.001$), except CD40L in HZ/su recipients (Fig. 3). Compared to baseline, the frequency of gE-specific CD4[+] T-cells at day 30 after vaccination increased 2.7-fold for IFN-γ positive, 3.4-fold for IL-2 positive, 2.2-fold for TNF-α positive, 1.5-fold for CD40L positive subsets in the LZ901 group, and 1.8-fold, 2.7-fold, 2.1-fold and 1.0-fold in the HZ/su group, respectively (Fig. 3 and Supplementary Table 3). At day 30 after vaccination, the frequency of IFN-γ-expressing CD4[+] T cells was significantly higher in the LZ901 group than that in the HZ/su group, with a median (IQR) of 198.0 (148.0, 261.0) vs. 165.0 (114.0, 226.0) per 10[5] Peripheral blood mononuclear cells (PBMCs) ($p = 0.0007$). For CD4[+] T-cell expressing cytokines IL-2 and TNFα, there were no significant differences between the two groups (på 0.05) (Fig. 3 and Supplementary Table 3).

In addition, we also observed a significant increase in gE-specific CD8[+] T-cell producing cytokines in LZ901 recipients at 30 days after the second dose, with a fold increase of 1.9 for IFN-γ positive, 2.6 for IL-2 positive, 1.4 for CD40L positive subsets compared to the baseline, respectively (all p < 0.05), but not for TNF-α. However, only significant increase was observed for IL-2 expressing -CD8[+] T-cell responses in the HZ/su group, with a fold increase of 1.1 compared to baseline (p = 0.0024) (Fig. 4). The frequency of CD8[+] T-cell produced cytokines was significantly higher in the LZ901 group than those in the HZ/su group, with a median (IQR) of 347.0 (222.0, 495.0) vs 213.0 (154.5, 328.5) for IFN-γ (p < 0.0001), 235.0 (140.0, 376.0) vs 136.5 (83.5, 204.0) for IL-2 (p < 0.0001) and 92.0 (46.0, 165.0) vs 76.5 (38.0, 132.0) for CD40L subsets (p = 0.0298) per 10[5] PBMCs, respectively. While there was no significant difference for TNF-α-expressing CD8[+] T-cell between the two groups (på 0.05) (Fig. 4 and Supplementary Table 4).

Prior to vaccination, all participants were seropositive for anti-gE antibodies at baseline, with geometric mean concentrations (GMCs) of 1187.0 mIU/mL in LZ901 group and 1284.8 mIU/mL in HZ/su group, respectively. Both LZ901 and HZ/su vaccines induced significant increases in terms of anti-gE antibodies at 30 days after the second dose, with the geometric mean fold increase (GMFI) of LZ901 group and HZ/su group were 24.4 (95%CI: 20.9, 28.3) and 51.9 (44.0, 61.3). However, the post-vaccination anti-gE antibody concentrations were significantly lower in LZ901 recipients than those in HZ/su recipients (p < 0.0001), with the GMCs of 28930.6 mIU/mL and 66758.5 mIU/mL, respectively. Compared to baseline before vaccination, 98.6% (139/141) of participants seroconverted for anti-gE antibody at 30 days after receiving two doses of LZ901 vaccine, while all participants seroconverted in HZ/su group (Table 3 and Supplementary Fig. 1).

In LZ901 participants, post-vaccination CD4[+] T-cell responses secreting IFN-γ, IL-2, and TNF-α exhibited significant weak correlations with anti-gE antibodies at 30 days after vaccination (IFN-γ: r = 0.174, p = 0.0385; IL-2: r = 0.185, p = 0.0277; TNF-α: r = 0.293, p = 0.0004). There was no correlation between post-vaccination anti-gE antibody levels and cell mediated immunity (CMI) responses at day 30 for HZ/su group (Supplementary Fig. 2). Multivariable analysis revealed the baseline immunity to VZV strongly affected the post-vaccination responses in both vaccine groups, showing higher anti-gE antibodies pre-vaccination correlated with elevated antibodies 30 days post-vaccination, while higher levels of baseline frequencies of cytokine-producing CD4[+] or CD8[+] T cells correlated with higher corresponding post-vaccination CMI responses. While age and sex had no significant impact on anti-gE antibody and CMI responses post-vaccination (p > 0.05 for all comparisons) (Supplementary Table 7).

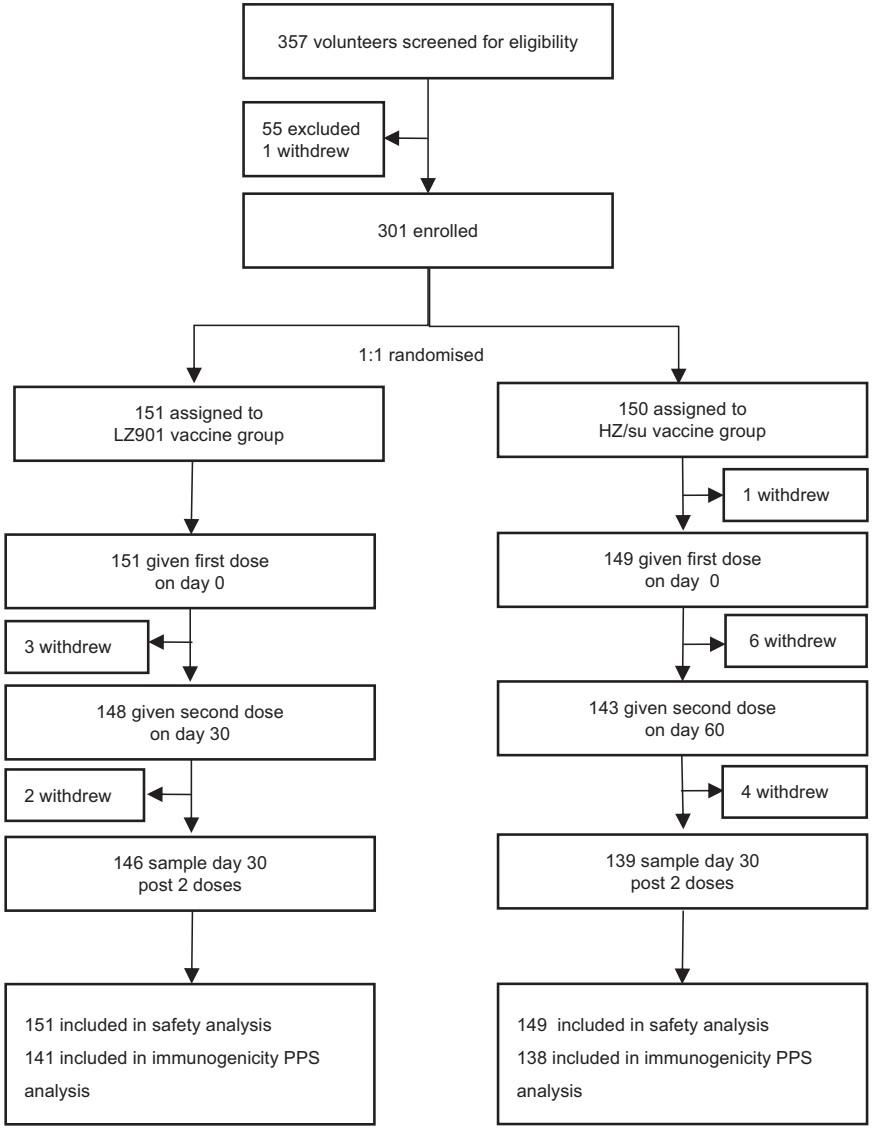

**Fig. 1 | Trial profile.** The immunogenicity PPS analysis set included all participants who received two doses within the preset time window, had no major protocol violations, and had pre- and post-vaccination immunogenicity data available. 6 participants were excluded from immunogenicity PPS analysis set because of the second vaccination time beyond the time window, with 5 in LZ901 group and 1 in HZ/su group, see supplementary table 9. PPS, per-protocol set.

## Safety

Within 30 days after each dose, 41.1% (62/151) of participants in the LZ901 vaccine group reported at least one adverse reaction, which was significantly lower than that (87.9%, 131/149) reported in the HZ/su vaccine group (p < 0.0001) (Table 4). Injection-site solicited reactions occurred in 27.2% (41/151) of LZ901 recipients and in 82.6% (123/149) of HZ/su recipients (p < 0.0001), respectively. The most common injection site adverse reactions were pain (80.5% [120/149] vs. 27.2% [41/151]), redness (16.8% [25/149] vs. 0.7% [1/151]), swelling (16.1% [24/149] vs. 2.7% [4/151]) and pruritus (14.8% [22/149] vs. 0), which were all more frequently reported in the HZ/su vaccine group than the LZ901 vaccine group did. Systemic solicited reactions were reported by 14.6% (22/151) of LZ901 recipients and 54.4% (81/149) of HZ/su recipients (p < 0.0001). The most common systemic reactions were fever (42.9% [64/149]), weakness (27.5% [41/149]), headache (13.4% [20/149]), muscle pain (12.1% [18/149]) and fatigue (11.4% [17/149]) in the HZ/su group, While the frequences of these systemic reactions were low in the LZ901 group.

Most of the reported adverse reactions were mild or moderate in severity. Grade 3 adverse reactions were only reported by one of LZ901 recipients and nine of HZ/su recipients. A slightly lower proportion of participants reported solicited reactions after the second dose compared to the first dose (15.5% vs. 27.1% in LZ9001 recipients; 69.9% vs. 77.8% in HZ/su recipients) (Supplementary Fig. 3). The occurrences of unsolicited reactions were similar between the two groups (11.9% [18/151] in the LZ901 group vs 12.8% [19/149] in the HZ/su group). Eight participants reported serious adverse events (SAEs), with four in the LZ901 group and four in the HZ/su group, within 30 days post-vaccination. However, none of the SAEs were judged to be causally related to the vaccination (Supplementary Table 8).

## Discussion

This randomized trial provides the first head-to-head comparison of the immunogenicity and safety profiles of the gE-Fcs fusion protein subunit vaccine candidate LZ901 and the licensed adjuvanted subunit vaccine (HZ/su) for HZ prevention. LZ901 is a HZ vaccine candidate in development, contains 100 µg of tetrameric recombinant VZV gE-Fcs and aluminum hydroxide adjuvant per 0.5 ml in each dose. While, HZ/su vaccine contains 50 µg of recombinant VZV gE and the liposome-based AS01B adjuvant system per 0.5 ml in each dose. HZ/su had extremely high protective effects, with å 90% effectiveness against HZ

and PHN in elderly people aged ≥50 years[7,8,11], establishing itself as the benchmark for next-generation zoster vaccines. A head-to-head immunogenicity comparison would be helpful to infer the relative protective potential of LZ901 vaccine containing the same antigens as Hz/su.

Our findings demonstrate that LZ901 elicits non-inferior CD4[+]/CD8[+] T-cell responses compared to the licensed HZ/su vaccine. Although both vaccines elicited robust CD4[+] T-cell responses against VZV gE, LZ901 showed significantly higher IFN-γ[+] CD4[+] T-cell expansion compared to Hz/su (2.7-fold vs 1.8-fold increase over baseline) and a greater proportion of CD4[2+] T-cell responders in LZ901 recipients (83.0% vs. 58.1%). Notably, LZ901 generated higher CD8[+] T-cell responses than HZ/su, with the proportion of CD8[2+] T-cell responders 46.8% vs. 8.8%. Interestingly, LZ901 induced lower levels of anti-gE antibody compared to HZ/su, despite its higher antigen dose, with GMC of 28930.6 mIU/mL versus 66758.5 mIU/mL.

The superior cellular immunity but lower humoral responses observed with LZ901 compared to HZ/su can be attributed to fundamental differences in their antigen design, adjuvant systems, and immune activation pathways. LZ901's gE-Fc fusion structure engages FcγR on antigen-presenting cells, facilitating efficient antigen inter-nalization and cross-presentation via MHC class I and II pathways[12,13], which are critical for eliminating virus-infected cells and controlling viral reactivation. In addition to providing T cells with co-stimulatory signals via cell surface molecules, FcγR engagement can also trigger the release of soluble factors, such as cytokines, that can enhance T-cell activation[14,15]. In contrast, HZ/su's non-fused gE antigen, combined with the AS01B adjuvant, primarily enhances MHC class II presentation and Th1/Th2-balanced CD4[+] T-cell responses[16]. Aluminum hydroxide (used in LZ901) is a Th2-skewed adjuvant that promotes antibody class-switching but is less effective at generating high-affinity, long-lived plasma cells[17]. AS01B adjuvant contains the toll-like receptor 4 agonist MPL, which is known to stimulate B-cell help through follicular helper T cells in draining lymph nodes, enhances high-titer antibody production[18,19]. This likely explains why LZ901 induced superior cellular immunity but lower humoral responses compared to HZ/su. While antibodies play a role in neutralizing extracellular virus and mediating antibody-dependent cellular cytotoxicity[20,21], CMI is the primary correlate of protection against HZ, as it directly targets latently infected neurons and prevents viral reactivation[22,23]. The heightened CMI induced by LZ901 could compensate for the effect of lower antibodies on the vaccine efficacy. Ongoing phase 3 trials will clarify whether LZ901's CMI-driven profile translates to clinical efficacy comparable to HZ/su.

Antibody and CMI magnitudes both were correlated with clinical protection against HZ, the correlation between these two types of indicators is uncertain. For live attenuated vaccine Zostavax, some studies reported weak correlations and others exhibited no significant antibody-CMI correlation[24,25]. While the adjuvanted subunit vaccine HZ/su showed transient moderate correlations between antibody and CMI[26]. In our study, significant but weak correlations between antibody (including antibody fold-rise) and CMI responses were observed in LZ901 recipients. However, there was no significant correlation between the two indicators for HZ/su vaccine. Nevertheless, antibody and antibody fold-rise are more likely to be a nonmechanistic correlate of the protective immunity against HZ, indicating responsiveness to HZ vaccine.

The increased incidence of HZ in older adults is believed to be linked to an age-related decline in cell-mediated immunity to VZV[5,27]. Aging leads to a gradual dysregulation of immune functions, resulting in reduced vaccine responses. HZ/su was by far the most efficacious for combating immunosenescence in older individuals, with proven high efficacy against HZ and PHN in all age groups, including those 80 or older. In our study, immune responses to the LZ901 vaccine exhibited minimal changes with increasing age. Similar to HZ/su, in LZ901 recipients, CD4[+]/CD8[+] T-cell and antibody responses following two doses of vaccine were similar in the recipients at 50–59 years, 60–69 years,

## Table 1 | Baseline characters of the participants

| | LZ901 group (N = 151) | HZ/su group (N = 149) | Total (N = 300) |
|---|---|---|---|
| Age, years | 60.5 (7.3) | 59.8 (7.1) | 60.2 (7.2) |
| Age group, n (%) | | | |
| 50–59 years | 75 (49.7) | 77 (51.7) | 152 (50.7) |
| 60–69 years | 55 (36.4) | 59 (39.6) | 114 (38.0) |
| ≥70 years | 21 (13.9) | 13 (8.7) | 34 (11.3) |
| Sex, n (%) | | | |
| Male | 65 (43.0) | 55 (36.9) | 120 (40.0) |
| Female | 86 (57.0) | 94 (63.1) | 180 (60.0) |
| Ethnicity, n (%) | | | |
| Han ethnic | 150 (99.3) | 146 (98.0) | 296 (98.7) |
| Others | 1 (0.7) | 3 (2.0) | 4 (1.3) |
| BMI, kg/m² | 24.6 (3.3) | 24.9 (3.3) | 24.7 (3.3) |
| Comorbidities diseases, n (%) | | | |
| Hypertension | 43 (28.5) | 38 (25.5) | 81 (27.0) |
| Diabetes | 13 (8.6) | 16 (10.7) | 29 (9.7) |

Data are mean ± SD or n (%). n = number of participants. % = proportion of participants. SD = Standard deviation.

## Table 2 | The proportion of participants with simultaneous positive responses to ≥2 cytokines

| | LZ901 group n/N (%) | HZ/su group n/N(%) | Difference % (95% CI) (LZ901 vs. HZ/su) | p value |
|---|---|---|---|---|
| The proportion of CD4[2+] T-cell responders | | | | |
| All age | 117/141 (83.0) | 79/136 (58.1) | 24.9 (14.5,35.3) | <0.0001 |
| 50–59 years | 57/73 (78.1) | 42/73 (57.5) | 20.6 (5.8,34.5) | <0.0001 |
| 60–69 years | 47/52 (90.4) | 29/50 (58.0) | 32.4 (12.6,52.2) | <0.0001 |
| ≥70 years | 13/16 (81.3) | 8/13 (61.5) | 32.6 (−12.9,52.3) | 0.03719 |
| The proportion of CD8[2+] T-cell responders | | | | |
| All age | 66/141 (46.8) | 12/136 (8.8) | 38.0 (28.5,47.5) | <0.0001 |
| 50–59 years | 36/73 (49.3) | 4/73 (5.5) | 43.8 (31.2,56.5) | <0.0001 |
| 60–69 years | 22/52 (42.3) | 6/50 (12.0) | 30.3 (14.1,46.5) | <0.0001 |
| ≥70 years | 8/16 (50.0) | 2/13 (15.4) | 34.6 (3.2,66.0) | 0.00267 |

N = the number of participants included in the analysis; n = the number of participants with simultaneous positive responses to ≥2 cytokines (referred to as gE-specific CD4[2+]/CD8[2+] T-cell responses; For each cytokine (IFN-γ, IL-2, TNF-α, or CD40L), a positive responder was defined as a 2-fold increase in cytokine-secreting T cells post-vaccination compared to pre-vaccination levels. In HZ/su group, there was one sample from each of 50–59 years and 60–69 years group that could not be tested due to insufficient cell quantity; 95% CIs of the rate difference were calculated using the Miettinen and Nurminen method, two-sided $\chi^2$ test or Fisher's exact test was used for comparisons between two groups.

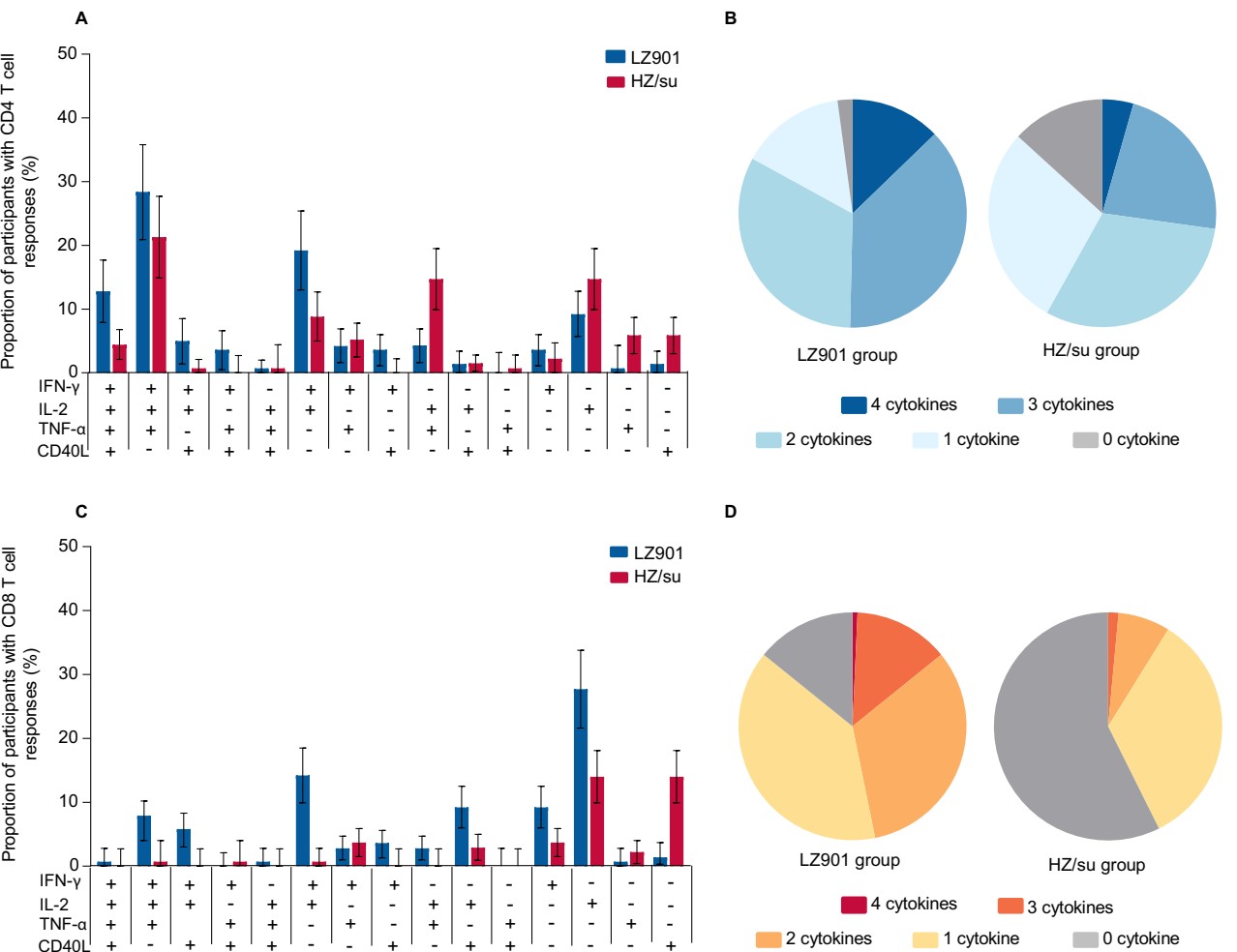

**Fig. 2 | The proportion of participants with simultaneous positive responses to any combination of cytokines. A**, **C** The proportion of participants with simultaneous positive responses to any combination of cytokines producing CD4 + T cells and CD8 + T cells in LZ901 group (n = 141) and HZ/su (n = 136) group. Error bars represent 95% confidence intervals estimated using the exact (Clopper–Pearson) method. **B**, **D** The proportion of participants with positive responses to only 1 cytokine and simultaneous positive responses to 2, 3, and 4 cytokines producing CD4 + T cells and CD8 + T cells in LZ901 group (n = 141) and HZ/su (n = 136) group. IFN-γ interferon-γ, IL-2 interleukin-2, TNF-α tumor necrosis factor-α, CD40L cluster of differentiation 40 ligand.

and ≥70 years of age. Considering that evidence has pointed to vaccine-induced CMI as being the primary mediators of protection against HZ, the preservation of CMI responses in older adults might help combat an age-related decrease in vaccine efficacy.

LZ901 exhibited lower reactogenicity than HZ/su (27.2% vs. 82.6% for injection-site reactions; 14.6% vs. 54.4% for systemic reactions), aligning with its aluminum hydroxide adjuvant's milder profile compared with AS01B adjuvant system. Although most adverse reactions were mild or moderate, the grade 3 adverse reactions were reported in 0.7% (1/151) of LZ901 recipients and 6.0% (9/149) of HZ/su recipients. The frequency and severity of the observed adverse reactions in the HZ/su recipients were in line with the results of previously reported studies with HZ/su vaccine[28]. There were no reports of vaccine-related SAEs or immune-mediated diseases for either vaccine during the study period. Overall, the safety and reactogenicity profiles of LZ901 were better than HZ/su.

There are several limitations to our study. First, the unblinded second-dose administration might introduce potential bias in safety evaluations for the second dose. Although the overall incidences of adverse events (AEs) following dose 2 were slightly lower in both groups compared to dose 1 (LZ901: 15.5% vs. 27.2%; HZ/su: 69.9% vs. 77.9%), the between-group differences in AE patterns for dose 2 remained consistent with those observed after dose 1 under blinded conditions. This consistency across blinded and unblinded phases strengthens the validity of our safety evaluation. Second, we only reported CMI and antibody responses within 30 days after vaccination, and did not report the durability of CMI and antibody responses over time of LZ901. Considering the long-term persistence of immunity is a key factor influencing vaccine effectiveness, a long-term immunogenicity follow-up is necessary. Third, only 34 (11.3%) adults aged over 70 years were enrolled in this study (21 in the LZ901 and 13 in the HZ/su vaccine groups). The small sample size of the elderly individuals warrants caution in extrapolating results to older adults, who generally face higher HZ risks and immunosenescence challenges. Fourth, the absence of unstimulated negative controls in the intracellular cytokine staining (ICS) assay precluded comprehensive background subtraction and restricted our ability to perform advanced polyfunctionality analyses using computational tools such as COMPASS[29]. While we validated the stability of background cytokine expression in the phase 1 trial of LZ901, showing no significant difference between pre- and post-vaccination unstimulated samples, the lack of unstimulated negative controls could underestimate response rates for each cytokine. In addition, our study is a randomized controlled trial, so even if there is some assay batch effect, it would be balanced between the two groups. Finally, our analysis focused solely on T-cell responses and antibody levels

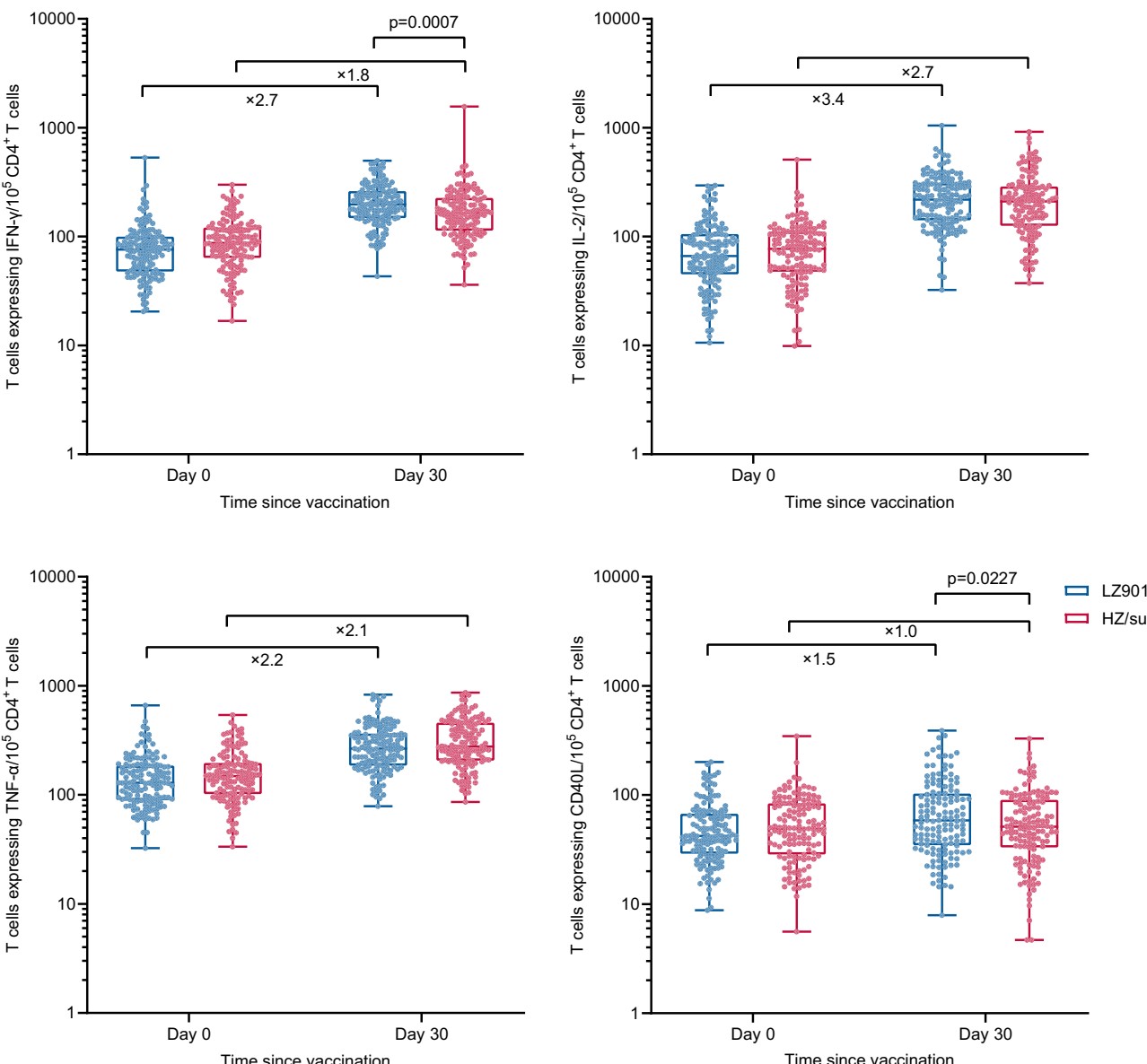

**Fig. 3 | The frequencies of cytokine-producing T cells per $10^5$ CD4$^+$ T cells.** The frequencies of cells secreting IFNγ, TNFα, IL-2, and CD40L from CD4$^+$ T cells in LZ901 group (n = 141) and HZ/su (n = 136) group. The box plots show the median (middle line) and the first and third quartiles (boxes), and the whiskers show the minimun to the maximum value. The numbers indicate the median fold of the frequencies of cytokine-producing CD4$^+$ T cells day 30 postvaccination to that baseline. Differences between groups were analyzed using a two-sided $t$ test after log-transformed T-cell frequency. Day 0, baseline before vaccination; Day 30, day 30 after the second dose. IFN-γ, interferon-γ; IL-2, interleukin-2; TNF-α, tumor necrosis factor-α; CD40L, cluster of differentiation 40 ligand.

between the LZ901 and HZ/su vaccines. The immunological mechanisms underlying the heterogeneity and homogeneity of T-cell and antibody immune responses between these two vaccines remain unknown. Further studies are essential for a deeper understanding of the observed differences between the two vaccines. With the advent of high-throughput "omics" technologies, systems immunology will provide significant insights into how two vaccines are modulated by the immune system to elicit protective immune responses.

In conclusion, two doses of LZ901 administered 30 days apart elicited robust cellular and humoral immune responses in adults aged ≥50 years, showed immunological non-inferiority to HZ/su in eliciting CD4 and CD8$^+$ T-cell responses, and a better safety profile. Although the protection of the LZ901 vaccine candidate against HZ remains uncertain, with similar CMI responses to HZ/su, better safety profile, and at a lower cost with adjuvant of aluminum

hydroxide, the LZ901 vaccine has the potential to be the optimal choice for national immunization programs, especially in LMICs. The ongoing multicenter, placebo-controlled phase 3 trial of LZ901 is expected to provide substantial evidence of the efficacy.

## Methods
### Study design and participants
This was a single-center, randomized, active-controlled, non-inferiority trial conducted in Wuxi, Jiangsu province, China. Eligible participants were healthy adults 50 years of age or older, and participants were excluded if they were previously vaccinated against varicella or HZ, had a history of HZ within 5 years before enrollment, allergic disease or reactions likely to be exacerbated by any component of the vaccine, had a confirmed or suspected immunosuppressive or immunodeficient condition. Detailed inclusion and exclusion criteria are included in the Supplementary Method.

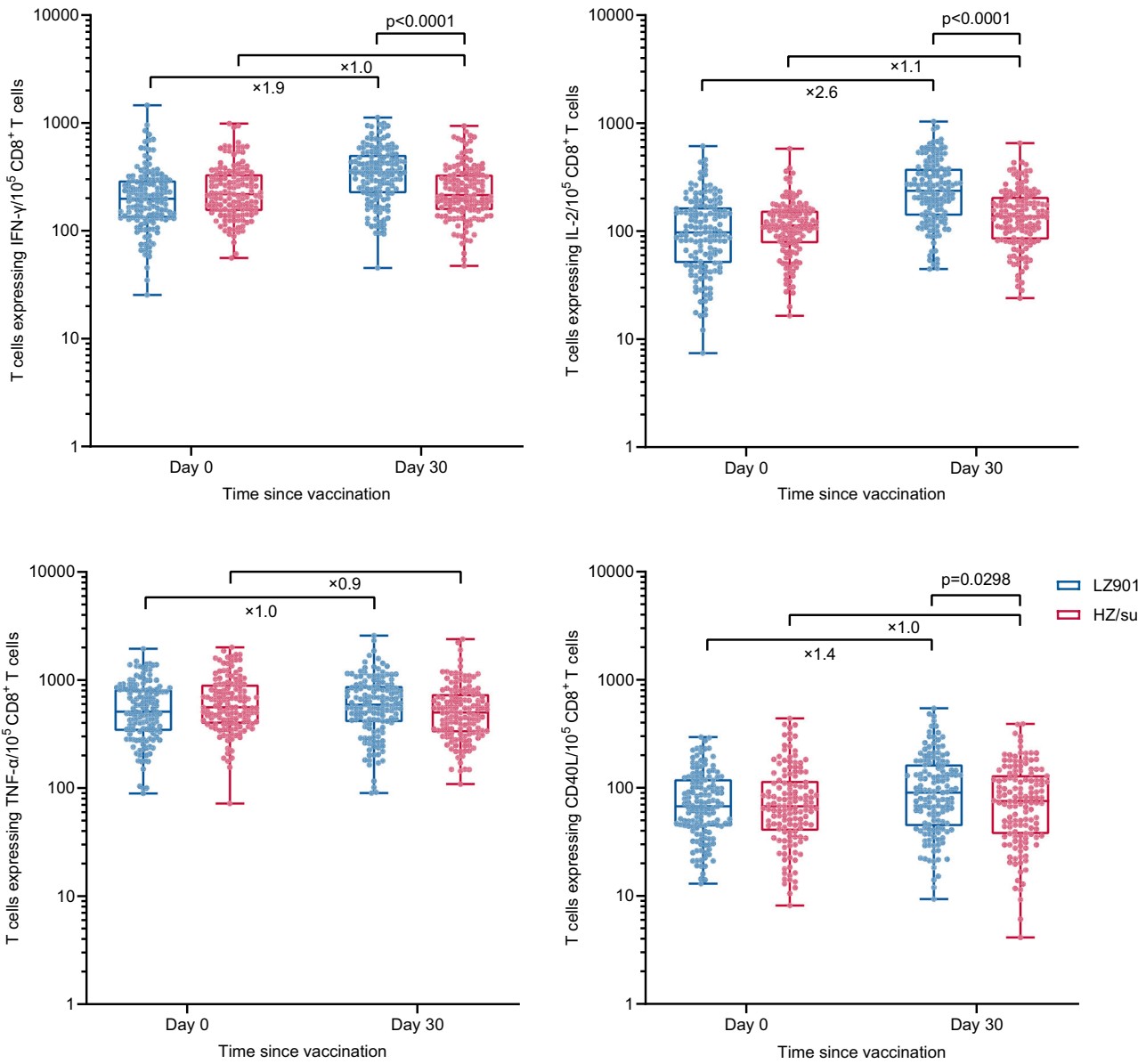

**Fig. 4 | The frequencies of cytokine-producing T cells per 10⁵ CD8⁺ T cells.** The frequencies of cells secreting IFNγ, TNFα, IL-2, and CD40L from CD8⁺ T cells in LZ901 group (n = 141) and HZ/su (n = 136) group. The box plots show the median (middle line) and the first and third quartiles (boxes), and the whiskers show the minimun to the maximum value. The numbers indicate the median fold of the frequencies of cytokine-producing CD8⁺ T cells day 30 postvaccination to that baseline. Differences between groups were analyzed using a two-sided $t$ test after log-transformed T-cell frequency. Day 0, baseline before vaccination; Day 30, day 30 after the second dose. IFN-γ, interferon-γ; IL-2, interleukin-2; TNF-α, tumor necrosis factor-α; CD40L, cluster of differentiation 40 ligand.

The protocol and informed consent were approved by the institutional review board of the Jiangsu Provincial Center of Disease Control and Prevention (the ethical approval number: JSJK2023-A021-01). Written informed consent from all participants was obtained before screening. The study was conducted in accordance with the Declaration of Helsinki and Good Clinical Practice guidelines. This study was registered at chictr.org (ChiCTR2300079076).

**Randomization and masking**
We used an interactive web-based response-randomization system for randomization. Eligible participants were randomly assigned in a 1:1 ratio to either the experimental vaccine LZ901 group or the active control vaccine HZ/su group. Randomization lists were generated by an independent statistician using SAS (version 9.4), with a block size of 4.

Due to the inherently different vaccination schedules between LZ901 (0–30 days) and HZ/su (0–60 days), blinding was maintained only for the first-dose safety assessment. Unblinded staff were designated for preparing vaccines out of sight of participants and other investigators, concealing the syringes with a label of randomization number, and administering vaccinations. Unblinded staff were aware of the treatment allocation, but were not allowed to be involved in any other trial procedures or to reveal this information to any participants or other investigators. Thus, all the other investigators maintained the blinding for the safety evaluation of the study from day 0 until 30 days following the first dose. And then, the group allocations were unblinded before the administration of the second dose. Therefore, both the participants and investigators, including safety assessors, were blinded for the AE assessments of the first dose, but were unblinded for that of the second dose. However, the laboratory personnel remained blinded

**Table 3 | Geometric mean concentrations and seroconversion rates of gE-specific IgG antibody according to age**

| | LZ901 group | HZ/su group | p value |
|---|---|---|---|
| **All ages** | | | |
| N | 141 | 138 | |
| GMC (Day 0) | 1187.0 (1022.0, 1378.7) | 1284.8 (1107.3–1490.8) | 0.459 |
| GMC (Day 30) | 28930.6 (26124.6, 32038.0) | 66758.5 (60815.8, 73281.8) | <0.0001 |
| Seroconversion rate (%) | 98.6 (95.0, 99.8) | 100 (97.4, 100.0) | 0.4983 |
| **50–59 years** | | | |
| N | 73 | 74 | |
| GMC (Day 0) | 1080.7 (886.5, 1317.6) | 1139.7 (938.9, 1383.4) | 0.7030 |
| GMC (Day 30) | 29488.9 (25898.9, 33576.6) | 63916.8 (56531.2, 72267.3) | <0.0001 |
| Seroconversion rate (%) | 100 (95.1, 100.0) | 100 (95.1, 100.0) | — |
| **60–69 years** | | | |
| N | 52 | 51 | |
| GMC (Day 0) | 1244.3 (951.3, 1627.6) | 1462.7 (1110.1, 1927.2) | 0.4008 |
| GMC (Day 30) | 28493.7 (24025.3, 33793.2) | 73540.8 (62316.1, 86787.4) | <0.0001 |
| Seroconversion rate (%) | 96.2 (86.8, 99.5) | 100 (93.0, 100.0) | 0.4951 |
| **≥70 years** | | | |
| N | 16 | 13 | |
| GMC (Day 0) | 1562.5 (974.3, 2505.8) | 1528.5 (1016.5, 2298.4) | 0.9419 |
| GMC (Day 30) | 27858.7 (17625.8, 44032.4) | 58504.3 (42535.4, 80468.3) | 0.0113 |
| Seroconversion rate (%) | 100 (79.4, 100.0) | 100 (75.3, 100.0) | — |

Data shown are the geometric mean (95% CI) for continuous variables and the percent (95% CI) for binary variables. $N$ = the number of participants included in the analysis. Seroconversion was defined as at least a fourfold increase in the antibody concentration after vaccination compared to the baseline level. For GMC, differences between groups were analyzed using a two-sided $t$ test after log-transformed antibodies. For seroconversion rate, 95% CIs of the rate were calculated using the Clopper-Pearson method, differences between groups were analyzed using a two-sided $\chi^2$ test or Fisher's exact test. GMC = geometric mean concentration.

throughout the trial. All samples were labeled with anonymized identifiers (participant ID + timepoint) and analyzed by an independent central laboratory with no access to randomization data or clinical records that could reveal group assignment.

**Procedure**

The experimental vaccine, LZ901, was developed by Beijing Luzhu Biotechnology Co., Ltd. contains 100 μg of tetrameric recombinant VZV gE-Fc protein and aluminum hydroxide adjuvant per 0.5 ml in each dose. While, HZ/su vaccine (Shingrix®, GlaxoSmithKline Biological SA) contains 50 μg of recombinant VZV gE and the liposome-based AS01B adjuvant system per 0.5 ml in each dose, was used as an active control. Both vaccines were stored at 2–8 °C, and administered (0.5 ml/dose) by intramuscular injection in the deltoid muscle in the upper arm. Eligible participants received two doses of vaccination with a 30 days interval for LZ901 vaccine and a 60 days interval for HZ/su vaccine between the two doses, respectively. The 30-day interval and 100 μg antigen dose for LZ901 were optimized in phase 1/2 trials. The safety of the high-dose group (100 μg per dose) showed no significant difference compared to the low-dose group (50 μg per dose), but the high-dose group demonstrated certain advantages in terms of anti-gE antibodies and cellular immunity.

Participants were asked to stay at least 30 min after each vaccination for any immediate adverse reactions and were instructed to record any solicited AEs up to day 7 after each vaccination on paper diary cards. Unsolicited AEs were recorded within 28 days after each dose, and SAEs were documented throughout the 6-month study period. AEs were graded according to the scale issued by the China State Food and Drug Administration (version 2019)[30]. The relationship of the AEs to the vaccination was made by investigators based on the WHO-UMC system for standardized case causality assessment[31].

Blood samples were collected for immunogenicity assessments at baseline before vaccination and day 30 after the second dose. PBMCs were collected from all participants to assess T-cell-mediated immune responses using ICS flow cytometry. Serum anti-gE antibody concentrations were measured by the China National Institutes for Food and Drug Control with an in-house enzyme-linked immunosorbent assay (recombinant gE was supplied by Beijing Luzhu Biotechnology Co., Ltd., Beijing, China). Details of immunological assay methods are provided in the Supplementary Method.

**Outcomes**

The primary outcomes was the proportion of participants with simultaneous positive responses to two or more cytokines (IFN-γ, IL-2, TNF-α, or CD40L) 30 days after the second dose (referred to as gE-specific CD4$^{2+}$/CD8$^{2+}$ T-cell responses). For each cytokine, a positive responder was defined as a 2-fold increase in cytokine-secreting T cells post-vaccination compared to pre-vaccination levels.

The secondary immunogenicity outcomes were the frequencies of cytokine-producing CD4$^+$/CD8$^+$ T cells per $10^5$ PBMCs, GMCs, GMFI, and seroconversion of gE-specific IgG antibodies day 30 after the second dose. Seropositivity of gE-specific IgG antibodies was defined as anti-gE antibodies concentration ≥100 milli-International Units (mIU)/mL. While the seroconversion was defined as a ≥4-fold increase in anti-gE antibodies concentration at day 30 compared to that at pre-vaccination. Safety outcomes included the incidence of adverse reactions within 30 days after each dose and SAEs over 6 months.

**Statistical analysis**

The phase 1 trial of LZ901 vaccine conducted in China, estimated that 85% of participants would have CD4$^{2+}$ T-cell responses and 55% would have CD8$^{2+}$ T-cell responses after LZ901 vaccination, compared to 75% and 45% for HZ/su participants. Assuming LZ901's T-cell responses are non-inferior to HZ/su, 150 participants per group are needed to achieve 90% power to show the lower limit of the 95% CI for the proportion difference is greater than −10%, with a one-sided $\alpha$ of 0.025, accounting for a 10% dropout rate. The calculation was performed by using Power Analysis and Sample Size software (version 11.0.7).

The primary immunogenicity analysis was performed in the per-protocol cohort consisted of all participants who met all the eligibility

**Table 4 | Solicited and unsolicited adverse reactions that occurred within 30 days after vaccination**

| | LZ901 (N = 151) | HZ/su (N = 149) | p value |
|---|---|---|---|
| All adverse reactions | | | |
| Any | 62 (41.1) | 131 (87.9) | <0.0001 |
| Grade 3 | 1 (0.7) | 9 (6.0) | 0.0101 |
| Local solicited adverse reactions | | | |
| Any | 41 (27.2) | 123 (82.6) | <0.0001 |
| Grade 3 | 0 | 2 (1.3) | 0.2458 |
| Pain | 41 (27.2) | 120 (80.5) | <0.0001 |
| Redness | 1 (0.7) | 25 (16.8) | <0.0001 |
| Grade 3 | 0 | 2 (1.3) | 0.2458 |
| Swelling | 4 (2.7) | 24 (16.1) | <0.0001 |
| Grade 3 | 0 | 2 (1.3) | 0.2458 |
| Pruritus | 0 | 22 (14.8) | <0.0001 |
| Induration | 0 | 7 (4.7) | 0.0069 |
| Systemic solicited adverse reactions | | | |
| Any | 22 (14.6) | 81 (54.4) | <0.0001 |
| Grade 3 | 0 | 8 (5.4) | 0.0034 |
| Fever | 2 (1.3) | 64 (42.9) | <0.0001 |
| Grade 3 | 0 | 5 (3.4) | 0.0292 |
| Weakness | 9 (5.9) | 41 (27.5) | <0.0001 |
| Grade 3 | 0 | 1 (0.7) | 0.4967 |
| Headache | 3 (1.9) | 20 (13.4) | 0.0002 |
| Muscle pain | 2 (1.3) | 18 (12.1) | 0.0002 |
| Fatigue | 3 (1.9) | 17 (11.4) | 0.0011 |
| Grade 3 | 0 | 1 (0.7) | 0.4967 |
| Insomnia | 3 (1.9) | 5 (3.4) | 0.4997 |
| Mucosal abnormality | 0 | 3 (2.0) | 0.1213 |
| Grade 3 | 0 | 1 (0.7) | 0.4967 |
| Pain at non-injection site | 3 (1.9) | 6 (4.0) | 0.3338 |
| Joint pain | 0 | 4 (2.7) | 0.0596 |
| Diarrhea | 4 (2.7) | 4 (2.7) | >0.9999 |
| Grade 3 | 0 | 1 (0.7) | 0.4967 |
| Vomiting | 1 (0.7) | 1 (0.7) | >0.9999 |
| Nausea | 2 (1.3) | 0 | 0.4984 |
| Unsolicited Adverse Reaction | | | |
| Any | 18 (11.9) | 19 (12.8) | 0.8267 |
| Grade 3 | 1 (0.7) | 1 (0.7) | >0.9999 |

Data are n (%). Any refers to all the participants with any grade adverse reactions. Two-sided $\chi^2$ test or Fisher's exact test was used for comparisons between two groups.

criteria, completed two-dose vaccination, and had pre- and post-vaccination immunogenicity data available. The safety analysis set consisted of all randomized participants who received at least one dose of LZ901 vaccine or HZ/su vaccine. The CMI responses were presented as the frequency of CD4$^+$ or CD8$^+$ T cells expressing each cytokine per $10^5$ PBMCs, and the number and proportion of participants with simultaneous positive responses to only 1 or any combination of 2, 3, or 4 cytokines. Medians with interquartile ranges were calculated for CD4$^+$ and CD8$^+$ T-cell frequencies. Two-sided 95% CIs of the proportion differences of CD4$^{2+}$ and CD8$^{2+}$ T-cell responders were computed using the Miettinen–Nurminen method. The antibodies were shown as GMCs with 95% CIs. We assessed the number and proportion of participants with adverse reactions post-vaccination and compared safety profiles between LZ901 vaccine and or HZ/su vaccine group. 95% CIs of the rate were calculated using the Clopper–Pearson method.

We analyzed categorical data with the $\chi^2$ test or Fisher's exact test, log-transformed antibodies and T-cell frequency with the $t$-test, and data that did not follow a normal distribution with the Wilcoxon rank-sum test. Spearman rank correlation coefficients were used to investigate the correlation between baseline and postvaccination gE-specific CMI responses with anti-gE antibodies. We used generalized linear regression models to identify independent factors associated with gE-specific IgG antibodies and CMI responses at 30 days after vaccination, involving variables including vaccine group, sex, age group (50–59 years, 60–69 years, and ≥70 years), pre-vaccination antibody levels, and pre-vaccination CMI responses. For co-primary endpoints (CD4$^{2}$+ and CD8$^{2}$+ T-cell response rates), a Bonferroni correction was pre-specified to control type I error at $\alpha = 0.0125$ (one-sided). Other reported $p$ values were two-sided with an $\alpha$ value of 0.05 unless otherwise specified. Statistical analysis was performed using SAS software (version 9.4) and R software (version 4.3.3). Flow cytometry data were analyzed with FlowJo X 10.0.7 R2 (Tree Star).

### Reporting summary
Further information on research design is available in the Nature Portfolio Reporting Summary linked to this article.

## Data availability
Individual participant data are available under restricted access for the requirements imposed by the Chinese Human Genetic Resources Administration concerning the public disclosure of clinical trial data. Researchers who provide a scientifically sound proposal will be allowed to access the de-identified individual participant data. Individual participant data can be obtained with a request to the corresponding author (jingxin42102209@126.com). All data generated in this study and the study protocol are provided in the Supplementary Information file.

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

## Acknowledgements

We thank all study participants enrolled in this trial. We gratefully acknowledge the participation and support of many staff in this study. This work was funded by the National Key Research and Development Program of China (grant number 2023YFC2307601, J.L.), Medical Research Program of Jiangsu Health Commission (grant number X20241509, P.J.), and Beijing Luzhu Biotechnology Co. The sponsor of the study participated in study design, but had no role in data collection, data analysis, data interpretation, or writing of the report.

## Author contributions

J.L. is the principal investigator of this trial. J.L., C.L., L.P., X.J., and P.J. designed the trials and the study protocol. P.J. drafted of the manuscript. J.L., C.L., F.Y., and S.T. contributed to critical review and revising of the report. P.J. and J.L. contributed to the data interpretation and revising of this manuscript. Y.Q., K.X., W.Y.W., and K.W. led the laboratory tests and analyses. F.Y., P.J., and X.C. were responsible for statistical analysis. H.P., W.J.W., and M.X. contributed to study supervision. S.X., Y.S., X.W., M.Y., K.Y., L.C., and A.Y. led and participated in the site work, including the recruitment, follow-up, and data collection. Y.X. and N.W. contributed to guide the processing of PBMCs. P.J. contributed to the literature search. L.P., J.K., and X.J. monitored the trial.

## Competing interests

J. K., X.J., and L.P. are employees of Beijing Luzhu Biotechnology Co., Ltd. All the other authors declare no competing interests.

## Additional information

**Peng-Fei Jin**[1,2,3,9], **Ya-Ru Quan**[4,9], **Shi-Xin Xiu**[5,9], **Xian-Min Jiang**[6], **Hong-Xing Pan** ®[2,3], **Yuan Shen**[5], **Xu-Wen Wang**[5], **Jian Kong**[6], **Wen-Juan Wang**[2], **Xiang Cao**[7], **Kang-Wei Xu**[4], **Min Yang**[5], **Kun Yang**[5], **Wen-Yan Wan** ®[4], **Kai-Qin Wang**[4], **Li Chen**[7], **Ai-Hua Yao**[7], **Yu-Peng Xue**[8], **Na Wan**[8], **Ming Xu**[2], **Shi-Yao Tao**[7], **Ling Peng** ®[6] ✉, **Fang-Rong Yan** ®[1] ✉, **Chang-Gui Li**[4] ✉ & **Jing-Xin Li** ®[1,2,3,7] ✉

[1]School of Science, Institute of Global Health and Emergency Pharmacy, China Pharmaceutical University, Nanjing, China. [2]NHC Key Laboratory of Enteric Pathogenic Microbiology, Engineering Research Center of Health Emergency, Jiangsu Province Center for Disease Control and Prevention, Nanjing, China. [3]School of Public Health, National Vaccine Innovation Platform, Nanjing Medical University, Nanjing, China. [4]National Institutes for Food and Drug Control, Beijing, China. [5]Wuxi Center for Disease Control and Prevention, Wuxi, China. [6]Beijing Luzhu Biotechnology Co. Ltd., Beijing, China. [7]Department of Endocrinology, School of Public Health, Zhongda Hospital, School of Medicine, Southeast University, Nanjing, China. [8]State Key Laboratory for Conservation and Utilization of Bio-resource and School of Life Sciences, Yunnan University, Kunming, China. [9]These authors contributed equally: Peng-Fei Jin, Ya-Ru Quan, Shi-Xin Xiu. ✉e-mail: pengling@luzhubiotech.com; f.r.yan@163.com; changguili@aliyun.com; jingxin42102209@126.com

