## [Peer Review file · Nature Communications]

Immunogenicity and safety of a Recombinant gE-Fc Fusion Protein Subunit Vaccine for Herpes Zoster in Adults ≥ 50 Years of Age: a randomised, active-controlled, non-inferiority trial

Corresponding Author: Professor Jing-Xin Li

Version 0:

Reviewer comments:

Reviewer #1

(Remarks to the Author)

We are pleased to review this promising candidate for a shingles vaccine. The study reports a well-designed randomized trial comparing the immunogenicity and safety of a novel gE-Fc fusion protein vaccine (LZ901) with the licensed HZ/su vaccine. The results suggest that LZ901 elicits non-inferior cellular immune responses and offers a better safety profile—an important finding for herpes zoster prevention, especially in older adults. While the methodology is generally sound and the statistical analyses appropriate, several aspects of the study design and reporting require clarification to enhance the robustness and transparency of the conclusions.

1. The manuscript mentions that ethical approval was obtained from the Ethics Committee of the Jiangsu Provincial Center for Disease Control and Prevention. However, the ethical approval number is not provided. It is recommended that the authors include this information. In addition, while adhering to the journal's formatting requirements, it may be more appropriate to move the clinical trial registration number (ChiCTR) to the study design section.
2. The investigational and control vaccines in this study follow different immunization schedules, making it difficult to maintain blinding—particularly for safety evaluations. Therefore, the description of the study as “partially blinded” (lines 53 and 141) may be imprecise. The authors are advised to clarify the measures taken to ensure the objectivity of evaluations, including any measures applied to adverse event assessment and laboratory testing.
3. There appears to be an imbalance in the distribution of participants aged ≥ 70 years between the two groups. The authors should explain why age group and sex were not used as stratification factors during randomization. Additionally, Figure 1 notes that five participants in the test group were excluded from the PPS due to delayed administration of the second dose beyond the allowed window—possibly all aged ≥ 70 . Could these delays be associated with adverse events? Also, were the immunogenicity responses of these five participants consistent with the overall PPS trend?
4. The authors report that the cellular immune responses were superior in the test group compared to the control group, while the humoral responses were lower. It is recommended to expand the discussion by incorporating immunological mechanisms to help interpret the potential implications of this response pattern for vaccine efficacy.
5. As the primary endpoint involves multiple cellular immune response indicators, the authors should address how type I error was controlled in the statistical analysis.

Reviewer #2

(Remarks to the Author)

Notes for Authors

Summary

L53 – “without access to the vaccines” is unusual wording.

Introduction

L95 – note that 90% efficacy is at 3+ years in the phase III study, efficacy falls later as you describe

L97 – “it is” instead of “which”

L146 – “allowed to be involved”

Methods

L196 – “it was estimated that 85% of participants would have CD4+ T cell responses and 55% would have with CD8+ T-cell responses after for LZ901 vaccine, respectively”

L171-173 – What was the viability of the cells used for CMI?

Discussion

L346 – The difference between the two vaccines was most striking for CD8+ cells. Given the nature of the immunogen, do you want to conjecture on the reason for this? You hint at this below (L361-370)

L353-358 – This text does not add much to the discussion.

L375-L378 – Any thoughts about why the antibody response was significantly lower (especially with more antigen)? You might comment on what the RZV adjuvant adds that aluminum hydroxide does not?

Could you mention why the second dose was given as 30 days and the gE dose was higher? I assume these decisions were based on pre-clinical or phase 1 studies.

Reviewer #3

(Remarks to the Author)

In this paper, authors studied immunogenicity data from a randomized, non-inferiority trial comparing a new vaccine candidate (LZ901) with HZ/su. The primary focus of the paper is to compare cellular immune responses, as quantified by percentage of gE-specific CD4+/CD8+ T cells expressing one or more cytokines. Adverse event rates and antibody data were also assessed and compared between two vaccines.

Overall, the paper is pretty clearly written; however, there are several places that require a higher degree of clarity and transparency.

First, it is not entirely clear how a participant was determined a 'vaccine responder.' This is arguably the most important message of the paper, so it requires greater clarity. I will break this down as follows.

1. After reviewing Table S1, it is clear that although the new vaccine candidate elicited more polyfunctional T cells, the active control and the new vaccine are comparable if we define a vaccine responder as showing ≥ 1 cytokine instead of ≥ 2 cytokines (141/141 vs 131/136). In other words, the main message of the paper need be qualified and make audience aware of the other side of the story.

2. Related to my last point. I would recommend authors consider using a polyfunctionality score, which is effectively a weighted sum of all functionalities, as an outcome. This will help address the issue I raised in my last point. A polyfunctionality score will give more weight to polyfunctional T cells and based on the current data, it appears that the new vaccine has an advantage from the polyfunctionality perspective. Here is a reference to constructing a score:
<https://www.nature.com/articles/nbt3187>

3. For a given cytokine combination, e.g., IFN γ and IL-2, authors stated in the footnote of Supp Table S1 that “an active response was defined as a statistically significant increase in the proportion of cytokine-secreting T cells postvaccination compared with that of pre-immunization.” It is relatively well-known that for ICS data, because the number of cells is large, a routine significance level of 0.05 often leads to poor type-I error rate control. Statistically, this is because the cells are correlated, so even though it appears we have 10K cells, the effective sample size is often much smaller than 10K. To remedy this, I would recommend authors using a tighter p-value threshold like 0.0001 when making a positive responder call for a cytokine combination. Alternatively, authors could show some negative control data and demonstrate the comparison of post- versus pre-vac cell proportion based on a routine two-sample test has correct type-I error control.

I also wonder if there is a control sample (unstimulated) in the ICS assay run.

The good thing is that, because the trial is randomized and the difference between two arms are fairly large, I expect the non-inferiority/superiority conclusion to hold for ≥ 2 cytokines.

Second, I wonder what is the percentage of CD4+ T cells expressing IFN γ and/or IL-2 in each arm.

Third, it is interesting that CMI and antibody are very weakly correlated. In previous works, it is shown that antibody foldrise and CMI are both correlated with clinical endpoint. I wonder if authors could report the correlation between CMI and antibody foldrise.

Finally, the data generated by the authors is a very good complement to the ongoing clinical trial. I look forward to seeing an analysis of immune correlates once the clinical outcomes are collected.

Version 1:

Reviewer comments:

Reviewer #1

(Remarks to the Author)

My previous questions have been well answered and I have no new comments.

Reviewer #3

(Remarks to the Author)

Authors responses are satisfactory. I just wanted to note one more thing. In the response letter, authors mentioned that the ICS assay did not use unstimulated controls. This is OK mostly because this is an RCT, so even if there is some assay batch effect, it would be balanced between two arms. This being said, I would expect authors to make this clearer in the paper.

REVIEWER COMMENTS

Reviewer #1 (Remarks to the Author):

We are pleased to review this promising candidate for a shingles vaccine. The study reports a well-designed randomized trial comparing the immunogenicity and safety of a novel gE-Fc fusion protein vaccine (LZ901) with the licensed HZ/su vaccine. The results suggest that LZ901 elicits non-inferior cellular immune responses and offers a better safety profile—an important finding for herpes zoster prevention, especially in older adults. While the methodology is generally sound and the statistical analyses appropriate, several aspects of the study design and reporting require clarification to enhance the robustness and transparency of the conclusions.

1. The manuscript mentions that ethical approval was obtained from the Ethics Committee of the Jiangsu Provincial Center for Disease Control and Prevention. However, the ethical approval number is not provided. It is recommended that the authors include this information. In addition, while adhering to the journal's formatting requirements, it may be more appropriate to move the clinical trial registration number (ChiCTR) to the study design section.

Response: Thank you. As your suggestions, we added the ethical approval number (JSJK2023-A021-01) and moved the clinical trial registration number (ChiCTR2300079076) to the study design section. Please see lines 337-341.

2. The investigational and control vaccines in this study follow different immunization schedules, making it difficult to maintain blinding—particularly for safety evaluations. Therefore, the description of the study as “partially blinded” (lines 53 and 141) may be imprecise. The authors are advised to clarify the measures taken to ensure the objectivity of evaluations, including any measures applied to adverse event assessment and laboratory testing.

Response: We agree that the differing immunization schedules (0-30 days for LZ901 vaccine vs. 0-60 days for the HZ/su vaccine) introduced challenges in maintaining blinding. In our study, unblinded staff were designated for preparing vaccines out of sight of participants and other investigators, concealing the syringes with a label of randomisation number, and administering vaccinations. Unblinded staff were aware of the treatment allocation, but were not allowed to involve in any other trial procedures or to reveal this information to any participants or other investigators. Thus, all the other investigators maintained the blinding for the safety evaluation of the study from day 0 until 30 days following the first dose. And then, the group allocations was unblinded

before the administration of the second dose. Therefore, both the participants and investigators including safety assessors were blinded for the adverse event (AE) assessments of the first dose, but were unblinded for that of the second dose. However, the laboratory personnel remained blinded throughout the trial. All samples were labeled with anonymized identifiers (participant ID + timepoint) and analyzed by an independent central laboratory with no access to randomization data or clinical records that could reveal group assignment. Please see line 348-363.

Nevertheless, we are aware that the unblinding adverse event assessment for the second dose might induce some bias. We have added it to the limitations in the discussion., please see lines 288-294:

” The unblinded second-dose administration might introduce potential bias in safety evaluations for the second dose. Although the overall incidences of AEs following dose 2 were slightly lower in both groups compared to dose 1 (LZ901: 15.5% vs. 27.2%; HZ/su: 69.9% vs. 77.9%), the between-group differences in AE patterns for dose 2 remained consistent with those observed after dose 1 under blinded conditions. This consistency across blinded and unblinded phases strengthens the validity of our safety evaluation.”

3. There appears to be an imbalance in the distribution of participants aged ≥ 70 years between the two groups. The authors should explain why age group and sex were not used as stratification factors during randomization. Additionally, Figure 1 notes that five participants in the test group were excluded from the PPS due to delayed administration of the second dose beyond the allowed window—possibly all aged ≥ 70 . Could these delays be associated with adverse events? Also, were the immunogenicity responses of these five participants consistent with the overall PPS trend?

Response: We appreciate the reviewer’s thorough evaluation and address the concerns as follows:

Age and sex showed no significant impact on anti-gE antibody and CMI responses post-vaccination in the previous phase 1/2 trials of the experimental vaccine LZ901, so stratification randomization by age and sex was not considered in this study. We employed block randomization with a block size of 4 to enhance balance between groups. Baseline analysis demonstrated comparable baseline characteristics between groups (Table 1), including sex distribution (the proportion of males: 43.0% VS 36.9%,

p=0.256) and age distribution (the proportion of the participants ≥ 70 years: 13.9% VS 8.7%, p=0.225).

Five participants in the experimental group were excluded from the PPS due to delayed administration of the second dose beyond the allowed window, of them 2 participants are aged ≥ 70 years. All five participants postponed their second dose due to acute upper respiratory infections during the winter influenza season. These acute upper respiratory infections were all considered as unrelated to the vaccination. Please see the Supplementary Table S9.

We added the full analysis set (FAS) analysis including the participants who were delayed for vaccination of the second dose. The results of the FAS analysis are consistent to the PPS analysis, please see the Supplementary Table S5 and S6.

4. The authors report that the cellular immune responses were superior in the test group compared to the control group, while the humoral responses were lower. It is recommended to expand the discussion by incorporating immunological mechanisms to help interpret the potential implications of this response pattern for vaccine efficacy.

Response: Thank you for your suggestion. We have expanded the Discussion section to incorporate immunological mechanisms that explain the observed divergence between cellular and humoral immune responses, as well as the potential implications for vaccine efficacy.

The superior cellular immunity but lower humoral responses observed with LZ901 compared to HZ/su can be attributed to fundamental differences in their antigen design, adjuvant systems, and immune activation pathways. Please see lines 232-255:

(1) Antigen Presentation and T-Cell Priming:

LZ901's gE-Fc fusion structure engages Fc γ receptors (Fc γ R) on antigen-presenting cells (APCs), facilitating efficient antigen internalization and cross-presentation via MHC class I pathways. This mechanism preferentially primes CD8⁺ cytotoxic T cells, which are critical for eliminating virus-infected cells and controlling viral reactivation. In addition to providing T cells with co-stimulatory signals via cell surface molecules, Fc γ R engagement can also trigger the release of soluble factors, such as cytokines, that can enhance T-cell activation. In contrast, HZ/su's non-fused gE antigen, combined with the AS01B adjuvant, primarily enhances MHC class II presentation and Th1/Th2-balanced CD4⁺ T-cell responses.

(2) Adjuvant-Driven Immune Polarization:

Aluminum hydroxide (used in LZ901) is a Th2-skewed adjuvant that promotes antibody class-switching but is less effective at generating high-affinity, long-lived plasma cells. AS01B adjuvant contains the toll-like receptor 4 agonist MPL, which is known to stimulate B-cell help through follicular helper T cells in draining lymph nodes, enhances high-titer antibody production. This explains why HZ/su achieved significantly higher anti-gE antibody levels despite its lower antigen dose.

While antibodies play a role in neutralizing extracellular virus and mediating antibody-dependent cellular cytotoxicity (ADCC), CMI is the primary correlate of protection against HZ, as it directly targets latently infected neurons and prevents viral reactivation. The ongoing phase 3 trials will clarify whether LZ901's CMI-driven profile translates to clinical efficacy.

5. As the primary endpoint involves multiple cellular immune response indicators, the authors should address how type I error was controlled in the statistical analysis.

Response:

In our study, the primary outcomes was the proportion of participants with simultaneous positive responses to two or more cytokines 30 days after the second dose (referred to as gE-specific CD4²⁺/CD8²⁺ T-cell responses). A positive responder was defined as 2-fold increase or more in cytokine-secreting T cells (IFN- γ , IL-2, TNF- α , or CD40L), post-vaccination compared to pre-vaccination levels. For co-primary endpoints (CD4²⁺ and CD8²⁺ T-cell response rates), a Bonferroni correction was pre-specified to control type I error at $\alpha = 0.0125$ (one-sided). Please see lines 447-449.

Reviewer #2 (Remarks to the Author):

Notes for Authors

Summary

L53 – “without access to the vaccines” is unusual wording.

Introduction

L95 – note that 90% efficacy is at 3+ years in the phase III study, efficacy falls later as you describe

L97 – “it is” instead of “which”

L146 – “allowed to be involved”

Methods

L196 – “it was estimated that 85% of participants would have CD4²⁺ T cell responses

and 55% would have with CD8²⁺ T-cell responses after for LZ901 vaccine, respectively”

Response: Thanks for your meticulous checking on the language. We have revised the sentence according to your suggestion.

L171-173 – What was the viability of the cells used for CMI?

Response: The viability of the cells used for CMI was more than 90%. We added the viability of the cells in experimental method section, please see supplementary Text S3.

Discussion

L346 – The difference between the two vaccines was most striking for CD8⁺ cells. Given the nature of the immunogen, do you want to conjecture on the reason for this? You hint at this below (L361-370)

Response: Thank you for your suggestions. We added the comment on the enhanced CD8⁺ T-cell response observed with LZ901 compared to HZ/su. Please see lines 232-243:

“The superior CD8⁺ T-cell response observed with LZ901 compared to HZ/su is likely due to its Fc-mediated antigen delivery mechanism. LZ901’s gE-Fc fusion structure engages Fcγ receptors (FcγR) on antigen-presenting cells (APCs), facilitating efficient antigen internalization and cross-presentation via MHC class I pathways. This mechanism preferentially primes CD8⁺ cytotoxic T cells, which are critical for eliminating virus-infected cells and controlling viral reactivation. In addition to providing T cells with co-stimulatory signals via cell surface molecules, FcγR engagement can also trigger the release of soluble factors, such as cytokines, that can enhance T-cell activation. In contrast, HZ/su’s non-fused gE antigen, combined with the AS01B adjuvant, primarily enhances MHC class II presentation and Th1/Th2-balanced CD4⁺ T-cell responses.”

L353-358 – This text does not add much to the discussion.

Response: We have deleted the sentence.

L375-L378 – Any thoughts about why the antibody response was significantly lower (especially with more antigen)? You might comment on what the RZV adjuvant adds that aluminum hydroxide does not?

Response: Thank you for your suggestions. We added the comment on why LZ901 induce lower anti-gE antibody titers despite its higher antigen dose. Please see lines 243-249:

“Aluminum hydroxide (used in LZ901) is a Th2-skewed adjuvant that promotes

antibody class-switching but is less effective at generating high-affinity, long-lived plasma cells. AS01B adjuvant contains the toll-like receptor 4 agonist MPL, which is known to stimulate B-cell help through follicular helper T cells in draining lymph nodes, enhances high-titer antibody production” This explains why HZ/su achieved significantly higher anti-gE antibody levels despite its lower antigen dose”.

Could you mention why the second dose was given as 30 days and the gE dose was higher? I assume these decisions were based on pre-clinical or phase 1 studies.

Response: Yes, the 30-day interval and 100 µg antigen dose for LZ901 were optimized in phase 1/2 trials. The safety of the high-dose group (100µg per dose) showed no significant difference compared to the low-dose group (50µg per dose), but the high-dose group demonstrated certain advantages in terms of anti-gE antibodies and cellular immunity. We added the basis for the immunization schedul and dosage selection in the Procedure section. Please see lines 374-378.

Reviewer #3 (Remarks to the Author):

In this paper, authors studied immunogenicity data from a randomized, non-inferiority trial comparing a new vaccine candidate (LZ901) with HZ/su. The primary focus of the paper is to compare cellular immune responses, as quantified by percentage of gE-specific CD4+/CD8+ T cells expressing one or more cytokines. Adverse event rates and antibody data were also assessed and compared between two vaccines.

Overall, the paper is pretty clearly written; however, there are several places that require a higher degree of clarity and transparency.

First, it is not entirely clear how a participant was determined a 'vaccine responder.' This is arguably the most important message of the paper, so it requires greater clarity. I will break this down as follows.

1. After reviewing Table S1, it is clear that although the new vaccine candidate elicited more polyfunctional T cells, the active control and the new vaccine are comparable if we define a vaccine responder as showing ≥ 1 cytokine instead of ≥ 2 cytokines (141/141 vs 131/136). In other words, the main message of the paper need be qualified and make audience aware of the other side of the story.
2. Related to my last point. I would recommend authors consider using a polyfunctionality score, which is effectively a weighted sum of all functionalities, as an outcome. This will help address the issue I raised in my last point. A polyfunctionality score will give more weight to polyfunctional T cells and based on the current data, it

appears that the new vaccine has an advantage from the polyfunctionality perspective. Here is a reference to constructing a score: <https://www.nature.com/articles/nbt.3187>

3. For a given cytokine combination, e.g., IFN γ and IL-2, authors stated in the footnote of Supp Table S1 that "an active response was defined as a statistically significant increase in the proportion of cytokine-secreting T cells postvaccination compared with that of pre-immunization." It is relatively well-known that for ICS data, because the number of cells is large, a routine significance level of 0.05 often leads to poor type-I error rate control. Statistically, this is because the cells are correlated, so even though it appears we have 10K cells, the effective sample size is often much smaller than 10K. To remedy this, I would recommend authors using a tighter p-value threshold like 0.0001 when making a positive responder call for a cytokine combination. Alternatively, authors could show some negative control data and demonstrate the comparison of post-versus pre-vac cell proportion based on a routine two-sample test has correct type-I error control.

4. I also wonder if there is a control sample (unstimulated) in the ICS assay run. The good thing is that, because the trial is randomized and the difference between two arms are fairly large, I expect the non-inferiority/superiority conclusion to hold for ≥ 2 cytokines.

Response: Thank you for your thorough and constructive suggestions. Below, we address each point raised and provide revisions to enhance transparency and robustness:

Response to comments 1 and 3:

We did not correctly describe the positive responder in the original version of the manuscript, and sorry for the confusion. We have corrected this content in the revised manuscript. For each cytokine (IFN- γ , IL-2, TNF- α , or CD40L), a positive responder was defined as a 2-fold increase or more in cytokine-secreting T cells post-vaccination compared to pre-vaccination levels. The primary outcomes was the proportion of participants with simultaneous positive responses to two or more cytokines 30 days after the second dose (referred to as gE-specific CD4²⁺/CD8²⁺ T-cell responses). Please see lines 398-402.

In addition, we analyzed the proportion of participants with positive responses to each cytokine, the results showed that LZ901 vaccine induced higher response rate to IFN- γ , IL-2 and at least one cytokine than HZ/su vaccine. Please see lines 129-134 and the supplementary table S1.

Response to comments 2 and 4:

We acknowledge the value of advanced polyfunctionality analysis. However, unstimulated controls were not included in the ICS assay, preventing COMPASS-based scoring analysis which requires background subtraction. The absence of unstimulated negative controls was based on the following considerations:

(1) Prior Validation of Background Stability

In LZ901 Phase 1 trial, comprehensive unstimulated control data confirmed that background cytokine frequencies were stable across pre- and post-vaccination, with no significant differences between pre- vs. post-vaccination backgrounds ($p > 0.05$).

(2) Primary Endpoint Design:

The study's primary immunogenicity outcomes are defined by fold-change from baseline, not absolute cytokine frequencies. This self-paired analysis inherently controls for individual background variation.

These validate that background noise does not materially impact CMI measured by fold-change of post-vaccination relative to pre-vaccination. Despite the absence of unstimulated controls, the higher proportions of LZ901 participants with simultaneous positive responses to ≥ 2 cytokines robustly demonstrate its superior CMI over HZ/su group. Please see supplementary table S2. We added the limitation in discussion section for the absence of unstimulated negative controls. Please see lines 302-308.

Second, I wonder what is the percentage of CD4+ T cells expressing IFN γ and/or IL-2 in each arm.

Response: We added the analysis for the proportion of participants with positive responses to each cytokine between two groups. The results showed that the proportion of participants with positive responses to IFN γ -producing CD4+ T cells in LZ901 and HZ/su group was 80.1%(113/141) and 42.6% (58/136), respectively, with 80.9% (114/141) and 66.9% (91/136) for the proportion of IL2-producing CD4+ T-cell responders. For the proportion of participants with simultaneous positive CD4+ T-cell responses to IFN γ and IL2, it was 65.2% (92/141) and 35.3% (48/136). Please see supplementary Tables S1 and S2.

Third, it is interesting that CMI and antibody are very weakly correlated. In previous works, it is shown that antibody foldrise and CMI are both correlated with clinical endpoint. I wonder if authors could report the correlation between CMI and antibody foldrise.

Finally, the data generated by the authors is a very good complement to the ongoing clinical trial. I look forward to seeing an analysis of immune correlates once the clinical outcomes are collected.

Response: Thank you. As your suggestions, we added the correlation analysis between antibody fold-rise and CMI in Supplementary Figure S2, which confirms a similarly weak correlation, consistent with the overall antibody-CMI relationship. We added the comment in discussion section, please see lines 256-266:

Although antibody and CMI magnitudes both were strongly correlated with clinical protection against HZ, the correlation between these two types of indicators is uncertain. For live attenuated vaccine Zostavax, some studies reported weak correlations and others exhibited no significant antibody-CMI correlation (J Infect Dis, 2013. doi: 10.1093/infdis/jit342; J Infect Dis, 2016. doi: 10.1093/infdis/jiv480). While the adjuvanted subunit vaccine HZ/su showed transient moderate correlations between antibody and CMI (J Infect Dis, 2018. doi: 10.1093/infdis/jiy095). In our study, significant but weak correlations between antibody (including antibody fold-rise) and CMI responses were observed in LZ901 recipients. However, there was no significant correlation between the two indicators for HZ/su vaccine. Nevertheless, antibody and antibody fold-rise are more likely to be a nonmechanistic correlate of the protective immunity against HZ, indicating responsiveness to HZ vaccine. The immunogenicity cohort of ongoing phase 3 efficacy trial of LZ901 will provide a change to conduct the correlate of protection analysis based on antibody responses and clinical outcomes.

Thank you again for the opportunity to improve our work. We hope these revisions meet your expectations.

REVIEWERS' COMMENTS

Reviewer #1 (Remarks to the Author):

My previous questions have been well answered and I have no new comments.

Response: Thank you again for the opportunity to improve our work.

Reviewer #3 (Remarks to the Author):

Authors responses are satisfactory. I just wanted to note one more thing. In the response letter, authors mentioned that the ICS assay did not use unstimulated controls. This is OK mostly because this is an RCT, so even if there is some assay batch effect, it would be balanced between two arms. This being said, I would expect authors to make this clearer in the paper.

Response: Thank you for your suggestion, we added the comment in the Discussion section. Thank you again for the opportunity to improve our work.

Fourth, the absence of unstimulated negative controls in the ICS assay precluded comprehensive background subtraction and restricted our ability to perform advanced polyfunctionality analyses using computational tools such as COMPASS29. While we validated the stability of background cytokine expression in phase 1 trial of LZ901, showing no significant difference between pre- and post-vaccination unstimulated samples, the lack of unstimulated negative controls could underestimate response rates for each cytokine. In addition, our study is a randomized controlled trial, so even if there is some assay batch effect, it would be balanced between two groups.